# Knee Joint Contact Forces during High-Risk Dynamic Tasks: 90° Change of Direction and Deceleration Movements

**DOI:** 10.3390/bioengineering10020179

**Published:** 2023-01-31

**Authors:** Giorgio Cassiolas, Stefano Di Paolo, Gregorio Marchiori, Alberto Grassi, Francesco Della Villa, Laura Bragonzoni, Andrea Visani, Gianluca Giavaresi, Milena Fini, Stefano Zaffagnini, Nicola Francesco Lopomo

**Affiliations:** 1Surgical Sciences and Technologies, IRCCS Istituto Ortopedico Rizzoli, 40136 Bologna, Italy; 2Department for Life Quality Studies, University of Bologna, 40136 Bologna, Italy; 32nd Orthopaedic and Traumatologic Clinic, IRCCS Istituto Ortopedico Rizzoli, 40136 Bologna, Italy; 4Education and Research Department, Isokinetic Medical Group, FIFA Medical Centre of Excellence, 40132 Bologna, Italy; 5Scientific Directorate, IRCCS Istituto Ortopedico Rizzoli, 40136 Bologna, Italy; 6Department of Information Engineering, University of Brescia, 25123 Brescia, Italy

**Keywords:** knee compartmental contact forces, musculoskeletal modeling, return to sport, anterior cruciate ligament, cut maneuver, deceleration, sport injury, football

## Abstract

Pivoting sports expose athletes to a high risk of knee injuries, mainly due to mechanical overloading of the joint which shatters overall tissue integrity. The present study explored the magnitude of tibiofemoral contact forces (TFCF) in high-risk dynamic tasks. A novel musculoskeletal model with modifiable frontal plane knee alignment was developed to estimate the total, medial, and lateral TFCF developed during vigorous activities. Thirty-one competitive soccer players performing deceleration and 90° sidestepping tasks were assessed via 3D motion analysis by using a marker-based optoelectronic system and TFCF were assessed via OpenSim software. Statistical parametric mapping was used to investigate the effect of frontal plane alignment, compartment laterality, and varus–valgus genu on TFCF. Further, in consideration of specific risk factors, sex influence was also assessed. A strong correlation (R = 0.71 ÷ 0.98, *p* < 0.001) was found between modification of compartmental forces and changes in frontal plane alignment. Medial and lateral TFCF were similar throughout most of the tasks with the exception of the initial phase, where the lateral compartment had to withstand to higher loadings (1.5 ÷ 3 BW higher, *p* = 0.010). Significant sex differences emerged in the late phase of the deceleration task. A comprehensive view of factors influencing the mediolateral distribution of TFCF would benefit knee injury prevention and rehabilitation in sport activities.

## 1. Introduction

The promotion of a physically active lifestyle is recognized as a primary tool for healthcare support and the prevention of harmful pathologies [1,2]. Health benefits related to sport practice across the lifespan include—but are not limited to—good cardiovascular functionality maintenance, osteogenic homeostasis preservation, weight management, psychological wellness, and social skills development [1,2,3,4,5,6]. Nevertheless, it is important to state that sports practice exposure is inextricably linked to the appearance of injuries [7], resulting in major risks of injury occurrence in members of the young population who have been increasingly dedicating themselves to intense physical activities [5,7]. Sport-related musculoskeletal injuries, such as anterior cruciate ligament (ACL) rupture, chondral damage, meniscal tears, and bone fractures, constitute a consistent burden for society. Injured individuals can require long rehabilitation periods or may even suffer long-term consequences regarding the health and function of articulating joints, as occurs in the case of early onset of post-traumatic knee osteoarthritis (OA) [8,9,10]. Traumatic joint injury has been recognized as a strong risk factor for the incidence of OA [11], as well as participation in highly dynamic sports where vigorous movements such as jumping, tackling, and pivoting maneuvers must be performed [11,12]. In particular, sharp deceleration and sidestepping maneuvers are among the most harmful movements for the integrity of knee articulation, leading to a major risk of tissue damage and ACL rupture [13,14].

Whilst participation in highly dynamic sports and traumatic joint injuries constitute modifiable risk factors in OA incidence, sex is an endogenous factor that has been proven to have an evidence-based influence on the occurrence of musculoskeletal injuries [15]. For example, female athletes are reported to be more likely to suffer bone stress injuries (in which higher rates of loading represent a risk factor) [15] and ACL tears (where tibiofemoral compression forces play a decisive role in combination with torque joint moments) [16,17]. Since overloading constitutes the primary mechanism underlying musculoskeletal injuries, a thorough understanding of how physical activities generate and shape joint contact forces according to boundary conditions would benefit the development of preventive and adaptive countermeasures for the prevention of burdens caused by injuries [18,19,20]. To date, knee tibiofemoral contact forces have been investigated mostly through running in a straight line [21,22,23] and jumping [24,25], while few studies have been conducted on sidestepping [21,26,27] and none on deceleration tasks. During the execution of the latter gestures, the knee joint has to support body weight and inertial forces through the generation of internal forces. Knee joint stability in these situations is ensured by the right amount of tibiofemoral compressive force [28], which is indeed indispensable for the good function of articulation. Frontal plane knee alignment plays a key role in determining the distribution of compressive forces on joint compartments and, consequently, the amount of injury risk; a more varus knee leads to higher forces in the medial compartment than the lateral one, and vice versa in the case of a valgus knee [29,30]. However, the effect of knee alignment on the distribution of joint compressive forces has been principally investigated in terms of gait [30,31], and no in-depth studies have been conducted on knee alignment during the performance of high-risk dynamic tasks.

A feasible and effective way to estimate tibiofemoral contact forces (TFCF) is through musculoskeletal models and biomechanical simulations. Musculoskeletal modeling allows for objective measurement of motion biomechanics, such as muscular activity and force generation [32,33,34], as well as measurement of their distribution throughout the human body system [21,30] in a non-invasive way. Several models have been proposed to investigate different targets in biomechanics, including either highly dynamic tasks such as running, jumping, cycling, or sidestepping [34,35] or the effect of joint misalignment [30]. Nevertheless, no studies have investigated joint reaction forces on medial and lateral compartments of the knee joint during high-risk dynamic tasks with models capable of personalizing varus–valgus alignment.

Therefore, the aim of the present study was to investigate (*i*) joint contact forces applied to the knee during deceleration and 90° change of direction tasks; (*ii*) their relationship with frontal knee alignment; and (*iii*) the presence of sex differences. We hypothesized that the exploitation of a generic musculoskeletal model with customizable varus–valgus knee alignment could better represent tibiofemoral loading distribution and could emphasize sexual and anatomical differences.

## 2. Materials and Methods

This is the secondary analysis of a larger validation study that aims to compare different methodologies (3D motion capture, wearable sensors, 2D video analysis) in the evaluation of a functional test protocol for return to sport after an ACL injury [36,37,38]. The functional tests were conducted at the Education and Research Department of the Isokinetic Medical Center of Bologna (Italy).

### 2.1. Participants

Overall, 34 recreational and elite athletes were recruited for the study. The inclusion criteria of the original study protocol were age between 18 and 50 years old and a Tegner activity level of at least 7. Exclusion criteria were the following: (*i*) evidence of musculoskeletal disorders or functional impairment; (*ii*) body mass index (BMI) > 35; (*iii*) previous surgery to lower limbs; (*iv*) cardiopulmonary or cardiovascular disorders; and (*v*) inability to perform the required tasks. All the subjects signed an informed consent form before starting the acquisition protocol. The research study was approved by the Institutional Review Board (IRB approval: 555/2018/Sper/IOR of 12/09/2018) of Area Vasta Emilia Romagna Centro (AVEC, Bologna, Italy) and was registered on ClinicalTrials.gov (Identifier: NCT03840551).

### 2.2. Data Collection

Each athlete performed a series of pre-planned 90° change of direction (COD) and frontal deceleration (DEC) tasks in a laboratory equipped with artificial turf. The COD task consisted of a frontal sprint followed by a 90° sidestep cut and a further frontal sprint in the new direction. The DEC task consisted of a frontal sprint followed by a sudden stop and backward sprint. Before the test, the subjects performed a 10 min dynamic warm-up and a few repetitions of the movements to build confidence with the environment and the motor task. Full foot contact on the force platform was required to consider the trial valid. All subjects performed three valid repetitions per lower limb.

The 3D motion analysis was recorded through a set of 10 stereophotogrammetric cameras (VICON Nexus, Vicon Motion Systems Ltd., Oxford, UK) and a force platform embedded in the floor (AMTI 400*600, Watertown, MA, USA). The systems were synchronized at a sampling frequency of 120 Hz.

System calibration was performed at the beginning of the acquisition and repeated periodically during the session. A total of 42 retroreflective markers were placed on each subject according to the full-body Plug-in-Gait protocol [36]. After marker positioning, model calibration for subjects was performed before each acquisition.

### 2.3. Data Processing

Marker trajectories were collected through the stereophotogrammetric cameras and ground reaction forces (GRF) were acquired through the force platform. Because of exporting issues, three individuals’ data were discarded. Ultimately, thirty-one trials were selected for the analysis (17 males, 14 females). To mitigate the effect of fatigue, the analysis considered the first repetition (i.e., trial) of the gesture performed with the preferred limb. Whenever the previous recording presented any issue, the recording that immediately followed was selected. Data were time-normalized to foot contact on the force platform using 5% of the GRF peak as a threshold for heel strike (0% of the movement) and toe-off (100% of the movement). Raw GRF were filtered using a zero-lag 2nd order Butterworth filter with 15 Hz cut-off frequencies.

### 2.4. Musculoskeletal Modeling

Estimation of kinematics, kinetics, muscular activation, and joint reaction forces was realized in OpenSim (v3.3, SimTK, Stanford, CA, USA) [32]. A validated generic model [35] was selected for the evaluation of high-risk dynamic tasks. Briefly, the model consisted of a whole-body model comprising 37 degrees of freedom. Specifically, it implemented ball-and-socket joints for the hips and revolute articulations for ankles, subtalar joints, and metatarsophalangeal joints. The knee was characterized by one degree of freedom in flexion, while the other degrees of freedom were flexion-dependent [39]. To allow for the evaluation of medial and lateral forces and the representation of non-neutral knee conditions, we also modified knee articulation architecture on the basis of Lerner et al.’s work [30]. We added two revolute frontal plane joints acting in parallel at the contact points of the medial and lateral compartments of each knee articulation, and the two frontal plane pin joints were articulated with femoral and tibial components, thus allowing frontal knee alignment to be tuned. For each subject, the generic model was scaled according to anthropometric measures, and varus/valgus alignment was determined on the basis of the orthostatic position of the joint during the static trial performed before acquiring the movements. In order to evaluate the effect of knee alignment personalization, two models for each individual were realized (Figure 1): the first with no frontal plane information (hereafter referred to as the “neutral knee model”) and the second with the application of varus or valgus modification (“aligned knee model”). Values of TFCF were normalized according to body weight (BW) to allow for inter-individual comparisons.

### 2.5. Statistical Analysis

Differences in subject characteristics between the male group and female group and between the varus genu group and valgus genu group were tested by using a Mann–Whitney U test for independent nonparametric samples (level of significance of α = 0.05).

The effect of variations in knee alignment on the distribution of compartmental tibiofemoral loads was evaluated through linear correlation (R, *p*-value). Specifically, changes in peak forces and task average forces due to angle correction were investigated. Statistical parametric mapping (SPM) [40] was exploited by using the SPM1D package (SPM, www.spm1d.org, v0.4 [41]) in MATLAB to evaluate statistically significant differences (α = 0.05) between (*i*) modeling approaches (i.e., neutral knee model and aligned knee model outputs), (*ii*) medial and lateral tibiofemoral contact forces, (*iii*) joint loadings in female and male athletes, and (*iv*) valgus and varus athletes. After verifying data demonstrated non-normal distribution, nonparametric tests were conducted. Specifically, two-tailed Wilcoxon signed-rank tests were exploited to analyze the differences between the modelling approaches and to highlight differences between medial and lateral tibiofemoral forces in aligned knee models, while nonparametric two-tailed Mann–Whitney U tests were performed on sex and varus–valgus investigations in aligned knee models. Differences were considered clinically relevant when statistically significant for at least 4% of the overall cycle [42].

## 3. Results

Overall, 31 valid trials from 31 participants were included in the analysis. The general characteristics of the subjects investigated are reported in Table 1. Body mass, height, and BMI were significantly different (*p* < 0.001) between the male and female groups; varus and valgus groups differed in terms of knee alignment (*p* < 0.001).

### 3.1. Change in TFCF According to Varus–Valgus Angle Correction

Regression analysis highlighted the presence of strong correlations between the magnitude of knee alignment modification and the relative change in force estimated by the model for both tasks. Regarding sidestepping, each degree of tibiofemoral angle in the varus direction increased mean medial force by 0.098 BW (R = 0.98, *p* < 0.001) and decreased mean lateral force by 0.116 BW (R = −0.98, *p* < 0.001) (Figure 2). Furthermore, each degree of tibiofemoral angle in the varus direction increased the medial peak by 0.133 BW (R = 0.91, *p* < 0.001) and decreased the lateral peak by 0.169 BW (R = −0.94, *p* < 0.001) (Figure 3).

Concerning the deceleration task, each degree of tibiofemoral angle in the varus direction increased mean medial force by 0.090 BW (R = 0.93, *p* < 0.001) and decreased mean lateral force by 0.103 BW (R = −0.98, *p* < 0.001) (Figure 2). Each degree of tibiofemoral angle in the varus direction increased the medial peak by 0.0843 BW (R = 0.71, *p* < 0.001) and decreased the lateral peak by 0.143 BW (R = −0.95, *p* < 0.001) (Figure 3).

### 3.2. Neutral vs. Aligned Models of TFCF

The total, medial, and lateral TFCF estimated by the neutral knee and aligned knee models were averaged considering all individual trials, and SPM analysis between the two modelling outputs was conducted. Regarding sidestepping, no remarkable differences were observed between the neutral and aligned knee models in terms of total contact force. However, medial loadings were significatively larger in the aligned knee models than in the neutral knee models (Figure 4) due to the fact that the varus alignment prevailed in the population. Conversely, lateral contact forces in aligned knee models were smaller than they were in neutral knee models throughout most of the tasks.

For the deceleration task, the aligned knee modeling approach presented similar loadings to the neutral counterpart in terms of total contact forces, statistically larger forces in the medial compartment at the beginning and end of the gesture, and smaller lateral compartment forces throughout the movement (Figure 5).

Due to the significant changes in modeling introduced by the customization of knee alignment, subsequent analyses considered only aligned knee modeling.

### 3.3. Medial vs. Lateral Compartment TFCF

Concerning sidestepping medial and lateral TFCF in aligned knee modeling, a maximal difference of nearly 3BW appeared during the initial part of the gesture, and according to SPM analysis, lateral TFCF were significatively higher than the medial forces at the beginning and end of the task (Figure 6).

Regarding the deceleration task, lateral TFCF were lower on average than medial forces throughout most of the gesture, except for the very first part of the movement where lateral forces were 1.5 BW higher. SPM highlighted significant differences at the beginning of the task and in the first half of the force plateau (Figure 7).

### 3.4. Male vs. Female TFCF

Focusing on sex differences related to the sidestepping activity, tibiofemoral forces related to female athletes’ activity were smaller than they were in males, although only a very small range of total contact force was significatively different in the second half of the movement. No remarkable differences were observed for medial and lateral contact forces (Figure 8).

Regarding the deceleration task, female athletes’ forces were similar to those of males until the final part of the movement, where significative differences were observed for total, medial, and lateral forces (Figure 9). A synoptic table with the average and peak forces for tibiofemoral compartments highlights the main differences between gestures and sex (Table 2).

### 3.5. Varus–Valgus Genu

No significant differences between the varus and valgus groups emerged from SPM analysis in relation to sidestepping and deceleration activities (Appendix A).

## 4. Discussion

In this study, we realized a multibody musculoskeletal model with customizable knee alignment to estimate tibiofemoral contact forces in young healthy athletes during 90° change of direction and deceleration tasks. First, we found that alignment personalization had a remarkable effect on estimated medial and lateral knee compartment loadings, consisting of a linear increase in medial contact forces and a decrease in lateral contact forces for each degree of varus modification. Second, we observed a nearly equal distribution of loadings on medial and lateral compartments during the execution of both tasks, except for the tendency of the lateral compartment to accept a major part of the loading during the initial braking phase. Third, we found no remarkable differences in loading between male and female groups during sidestepping, though female athletes presented smaller forces on average. conversely, during the final propulsive phase of the deceleration task, male athletes generated greater forces to a statistically significant degree. To our knowledge, this study was the first to explore the effect of frontal alignment on medial and lateral knee compartment reaction forces during the performance of highly dynamic tasks.

During sport practice, athletes may need vigorous physiological strategies to react to external stimuli. Deceleration and sidestepping activities are very common in sport—especially in team competitions—and can lead to injuries to the musculoskeletal system because of the high forces generated [43,44,45]. Both tasks share similar characteristics in that they present a first stage in which braking serves to decelerate body inertia, while in the second half of the tasks propulsive forces provide an acceleration of bodies in a perpendicular direction (90° sidestepping) or backwards (deceleration). Qualitatively, the analyzed gestures differed in terms of expressed maximal forces and loading curve shape. In fact, sidestepping leads to higher tibiofemoral loading values (up to 13 BW on average), specifically in concomitance with the middle part of the movement, where braking force for body deceleration partially overlaps with the propulsive force required for perpendicular acceleration. On the other hand, the deceleration gesture presents lower maximal forces (11.5 BW) and a flatter central region corresponding to the generation of static equilibrium forces; loading peak in deceleration should be searched at the beginning or at the end of the movement, where braking and propulsive forces are generated. Both knee compartments experience approximatively equivalent contact loads across both tasks. Concerning change of direction, this behavior was already reported in other studies [21,26,46] and was attributed to the generation of substantial muscle forces that contribute to knee stabilization during vigorous gestures. No previous studies have explored tibiofemoral forces in deceleration tasks. It has been reported that simpler gestures such as gait or straight running make the medial compartment bear most of the contact force [21,22,23]. In running, the medial tibiofemoral force is estimated to be nearly two-fold the lateral counterpart. In deceleration, the difference between loadings impacting on the two compartments was less evident, albeit the lateral compartment bore around one body weight less than the medial compartment throughout most of the movement. Regarding change of direction, compartmental forces overlapped for most of the movement.

Both deceleration and sidestepping presented major loading on the lateral side with respect to the medial side (1.5 ÷ 3 BW) during the very first part of the movement (Figure 6 and Figure 7). During sidestepping execution, contact loading would naturally shift from the medial to the lateral compartment because of the important external knee abduction moment generated by the vigorous movement [47]; however, this harmful behavior is prevented by the activation of muscles forces, which redistribute the total loading in approximately equivalent parts [21,27]. Nevertheless, our results seem to indicate that, in the first stages of braking, muscles are not ready to counterbalance the external moment, thus passing major bearing stresses to the lateral compartment. Interestingly, deceleration movement also showed a similar behavior, even if it was less pronounced. In this case, the reason beneath the major initial loading on the lateral side could depend on a medial shift of the knee during flexion, which thus generates an external abduction moment similar to the previous case (Appendix A). However, the initial part of braking was revealed as the most dangerous phase during both the considered movements. It is indeed the stage where lateral forces sharply rise and the major differences between compartments occur, which could be responsible for a loss of joint equilibrium and increased risk of injury [48].

The tibiofemoral contact forces estimated in this study were comparable to those reported in the literature. Studies implementing static optimization methods for musculoskeletal simulation reported loads higher than 12 BW for dynamic activities [49], whereas EMG-driven approaches implementing subject-specific muscular activations reported total tibiofemoral forces between 8 and 9 BW for <90° sidestepping tasks [26,46,50].

Frontal plane tibiofemoral realignment is known to alter the distribution of tibiofemoral contact forces between knee compartments, thus altering biological homeostasis [51] and affecting the risk of developing long-term diseases such as OA [52]. The presented knee model was revealed to be sensitive to varus–valgus alignment. An average increase in medial tibiofemoral force of 0.090 ÷ 0.098 BW per each degree of varus angle was observed throughout the simulations, whereas an average decrease in lateral loading (0.103 ÷ 0.116 BW per each varus degree) occurred. Similar behavior was observed for peak forces. This behavior is coherent with the knee model mechanism presented by Lerner et al. [30]. In that publication, a decrement of 0.078 BW in medial force and an augmentation of 0.045 in lateral force per each valgus angle was reported for the first peak of the gait task. The different magnitudes of loading correction may be due to both the different muscular architectures implementing the models and the different tasks involved in the simulations. In the latter case, a direct comparison to the same gesture could have provided better information.

No significant differences between male and female tibiofemoral compartmental forces emerged for sidestepping. It was highlighted that men generated on average one BW higher forces than women, though considering intra-population variance, task execution results were similar. Conversely, the presence of sex differences was highlighted in the last phase of the deceleration gesture, where a propulsive force is generated to start backwards movement. In this case, males generated 1.5 ÷ 2.5 BW higher forces than female athletes, revealing that male muscles provided more force when accomplishing this specific requirement. To the authors’ knowledge, no other studies have investigated loading differences between males and females performing deceleration or sidestepping tasks. However, the absence of differences in most of the compartmental loading curves between males and females observed in our study seems to agree with observations of OA incidence in the athletic population. A retrospective study evaluating the frequency and severity of MRI-based osteoarthritis in the main peripheral joints claimed that the knee is the most OA-affected articulation, with a similar incidence between male and female groups [53]. To investigate knee pathologies other than OA, estimating tibiofemoral contact forces alone is not enough to define a preventive risk. For example, ACL tears can occur due to multiple factors comprising—but not limited to—the magnitude of external forces, knee moments, kinematics, and hormonal conditions [54]. Higher forces on the lateral side, such as those highlighted in the initial braking phase of the present study, are linked to the concept of pivoting in sports; higher stress on the lateral side is usually counterbalanced by the action of the ACL. Thus, a particular attention to body deceleration stages should be considered to avoid undesired damage to ligaments.

According to the results of this work, no significant differences occurred between the varus and valgus groups. However, it must be highlighted that the two samples presented quite different prevalence (the valgus group comprised only seven individuals), which could have influenced the analysis. Despite this clear limit, it is worth mentioning that frontal knee alignment has to be considered as a main factor when determining mediolateral loading distribution deviations during simple tasks such as gait [29,55]. The above-presented SPM analysis highlighted a significant shift in forces due to varus/valgus condition during the execution of change of direction and deceleration tasks. From the perspective of personalized medicine, an adequate modification of knee alignment in concomitance with neuromuscular retraining related to the execution of tasks would be beneficial [52,56].

Several limitations affect this study. Firstly, we utilized an overall generic model, and this could be a simplification impacting on estimation of tibiofemoral contact forces as between-joint force development is sensitive to musculoskeletal geometries and architecture [57]. The effects of this limitation may be partially softened by the consistent number of individuals considered for the study [58] and the personalization provided by the implemented knee model, which is subject-specific in terms of frontal knee alignment. Nevertheless, this personalization strategy could also be questionable. In fact, the tibiofemoral varus–valgus alignment of subjects was estimated from a static trial recorded using an optoelectronic system; thus, in the end it was indirectly reconstructed by the position of external markers. Better information on knee alignment could undoubtably be achieved through medical imaging (i.e., magnetic resonance or radiographic images) of the articulation in orthostatic positions Moreover, this would have also provided the possibility of personalizing contact point locations (arbitrarily placed in the middle of condyles in this study), which has a role in the distribution of total joint loading across compartments [21,30,31]. However, alignment-informed models have proven to be more accurate in prediction than models that are only informed by contact points [30]; thus, that issue should only represent a minor weakness in the estimation of the tibiofemoral contact forces presented here. Moreover, the study lacks subject-specific information regarding muscular activity. As mentioned before, EMG-driven simulations have been shown to estimate lower total forces than general static optimization simulations. It is possible that the activation of specific muscular compartments could have had a non-negligible effect on the results. Finally, it has to be mentioned that the results presented here considered elite and recreational athletes together. Elite athletes, characterized in general by a higher amount of time afforded to training, could have received specialized training by team coaches with the aim of minimizing injury occurrence. However, the number of individuals included in the two groups was very dissimilar (27 elite athletes, 4 recreational athletes), so any statistical comparative analysis aiming to highlight dynamics differences lacked significance. Comparing differences in the tibiofemoral forces generated by elite and recreational groups with sex distinction would have provided a more comprehensive overview of task performance mechanisms.

## 5. Conclusions

In this study, we realized a multibody musculoskeletal model with customizable frontal knee alignment that was capable of estimating tibiofemoral contact forces during the execution of highly dynamic movements. This model was created with the aim of objectifying movement biomechanics to better address injury prevention and to support clinical practice in terms of rehabilitation.

Regarding the 90° change of direction and deceleration tasks, the results confirmed that a higher degree of varus knee is associated to larger medial contact forces. Inter-compartment forces were more similar in sidestepping than in deceleration, and the initial part of the movements provided the greatest risk of injury because of the inherent instability of the articulation. Since loading imbalance in knee compartments determines overstress in articular soft tissues (ligaments and cartilage), which could lead to serious injuries, specific attention should be paid to technique and rehabilitative exercises when training athletes’ neuromuscular response so that they are able to deal with the first moments of the deceleration and sidestepping braking phases.

As a further result of the study, we highlighted that, in general, men were prone to develop major internal forces during the execution of sports activities, though lateral forces were not higher in male individuals than female athletes, as the propension of valgus loading-related injuries in the latter group would seem indicate.

The biomechanical simulation of highly dynamic tasks, which often occur in game situations, is of paramount importance to understanding the underlying mechanisms of injury. Understanding the major contributors impacting on the execution of a certain task among all of the exogenous and endogenous factors would have the potential to illuminate specific training programs that could be used to prevent and reduce sport injury occurrence, thereby relieving the burden on national healthcare systems.

Future research should focus on developing more and more accurate tools to reliably estimate sport-related injury biomarkers. Additionally, exploiting novel technologies capable of recording motion outside of the lab would allow for a comprehensive evaluation of highly dynamic task situations. From this perspective, the recent trend of using inertial measurement units for motion analysis [59] could present interesting opportunities.

## Figures and Tables

**Figure 1 bioengineering-10-00179-f001:**
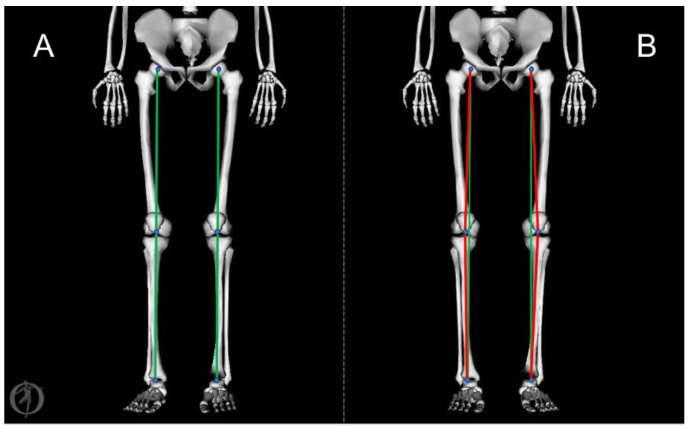
(**A**) Representation of the neutral knee model: hip, knee, and ankle joint centers are aligned along a straight line. (**B**) Representation of the aligned knee model (varus knees in the example above): hip, knee, and ankle joint centers are not aligned along a straight line.

**Figure 2 bioengineering-10-00179-f002:**
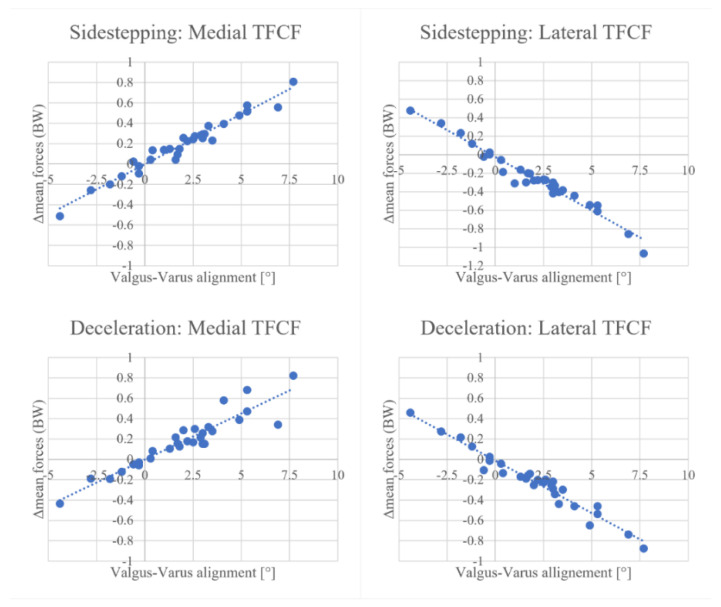
Linear correlations between valgus−varus alignment and mean force changes for medial and lateral tibiofemoral contact forces (TFCF) during sidestepping (**top row**) and deceleration (**bottom row**). Change (Δmean) is intended as the difference between task average forces measured by the aligned model with respect to the neutral model. Every dot represents one individual case.

**Figure 3 bioengineering-10-00179-f003:**
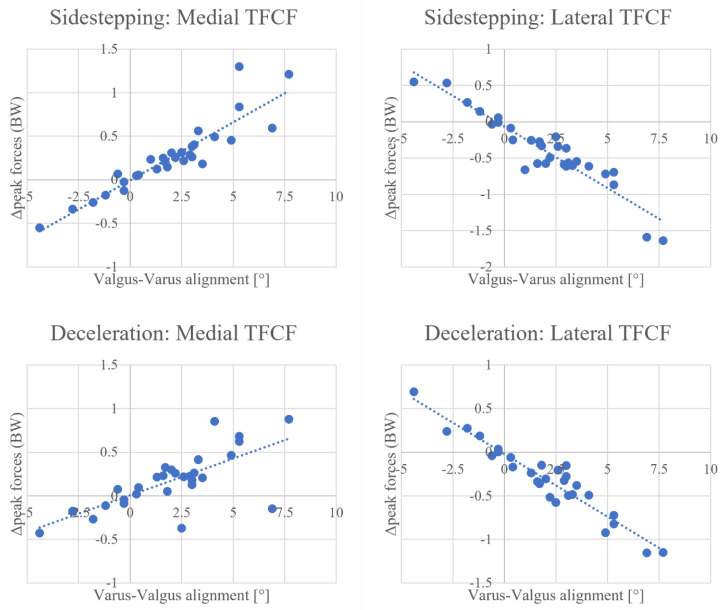
Linear correlations between valgus−varus alignment and peak force change for medial and lateral tibiofemoral contact forces (TFCF) during sidestepping (**top row**) and deceleration (**bottom row**). Change (Δmean) is intended as the difference between peak forces measured by the aligned model with respect to the neutral model. Every dot represents one individual case.

**Figure 4 bioengineering-10-00179-f004:**
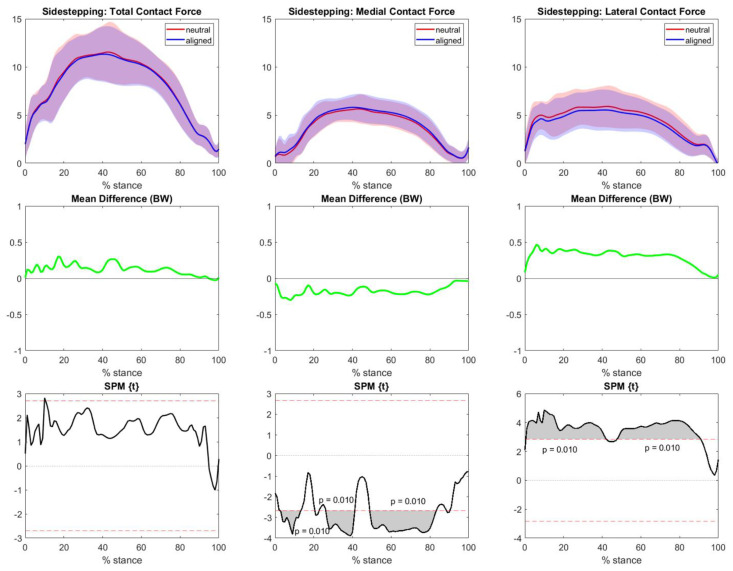
Sidestepping TFCF (normalized to body weight and represented as mean and standard deviation) of all individuals across the task stance in both the neutral (red) and aligned (blue) knee models. The left column represents total TFCF, the middle column represents medial TFCF, and the right column represents lateral TFCF. The central row highlights the differences between the means (neutral−aligned). Grey areas with corresponding *p*-values from SPM graphs indicate significant and clinically relevant differences.

**Figure 5 bioengineering-10-00179-f005:**
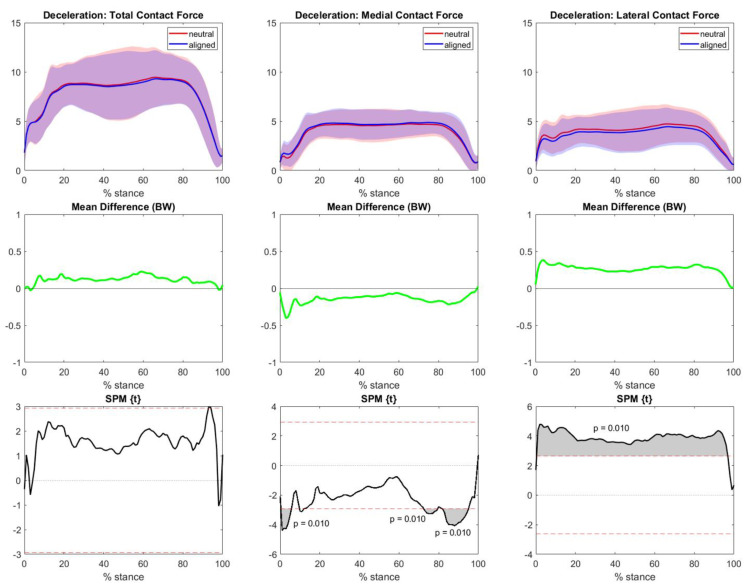
Deceleration TFCF (normalized to body weight and represented as mean and standard deviation) of all individuals across the task stance in both the neutral (red) and aligned (blue) knee models. The left column represents total TFCF, the middle column represents medial TFCF, and the right column represents lateral TFCF. The central row highlights the differences between the means (neutral−aligned). Grey areas with corresponding *p*-values from SPM graphs indicate significant and clinically relevant differences.

**Figure 6 bioengineering-10-00179-f006:**
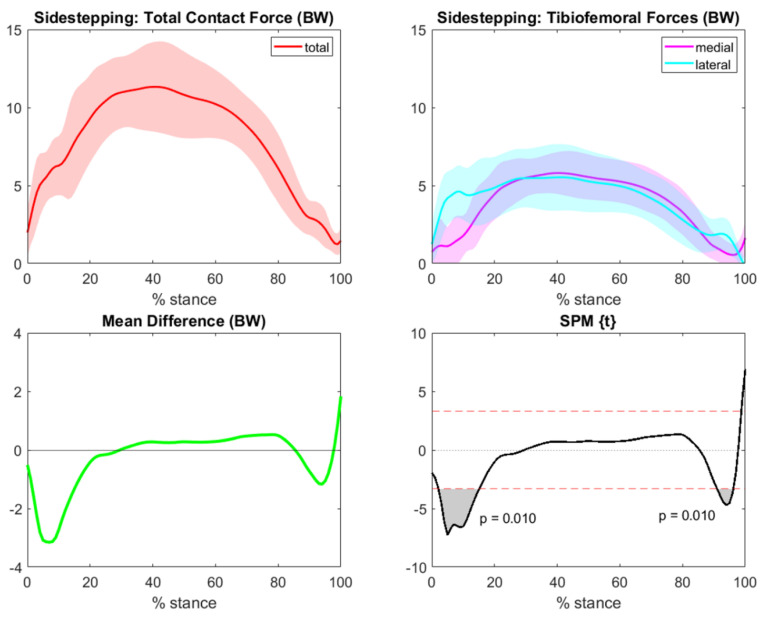
Total (red), medial (magenta), and lateral (cyan) sidestepping TFCF (normalized to body weight and represented as mean and standard deviation) across the task stance. The green line represents differences between means for the medial and lateral forces (medial−lateral). Grey areas with corresponding *p*-values from SPM graphs indicate significant and clinically relevant differences between compartmental forces.

**Figure 7 bioengineering-10-00179-f007:**
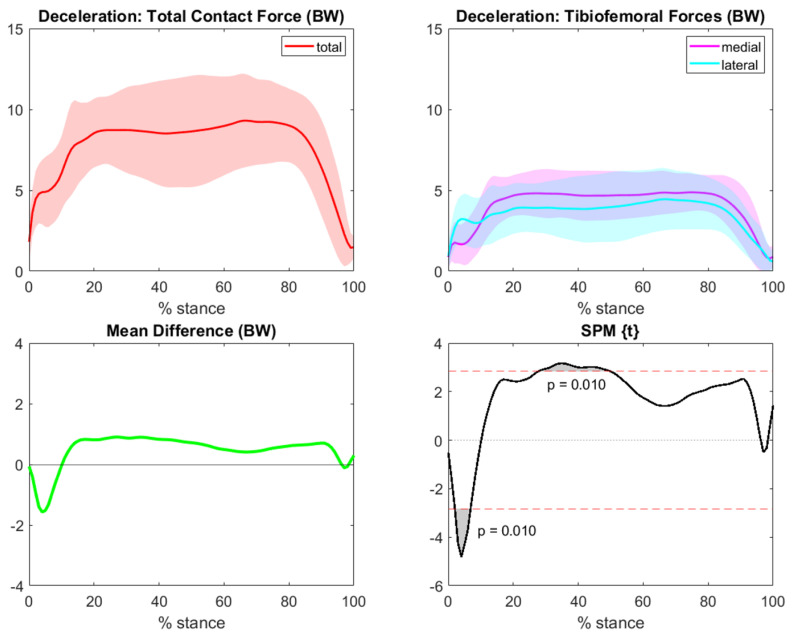
Total (red), medial (magenta), and lateral (cyan) deceleration TFCF (normalized to body weight and represented as mean and standard deviation) across the task stance. The green line represents differences between means for the medial and lateral forces (medial−lateral). Grey areas with corresponding *p*-values from SPM graphs indicate significant and clinically relevant differences between compartmental forces.

**Figure 8 bioengineering-10-00179-f008:**
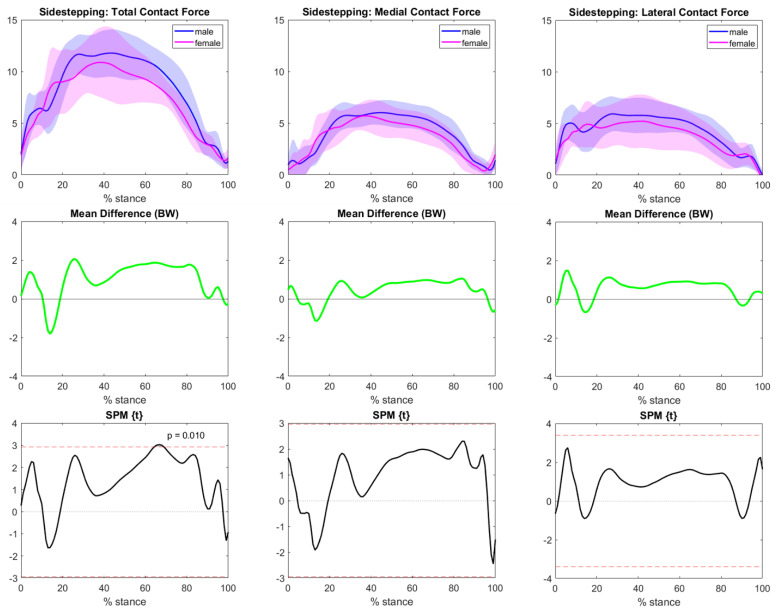
Sidestepping TFCF (normalized to body weight and represented as mean and standard deviation) across the task stance for both the male (blue) and female (magenta) groups. The left column represents total TFCF, the middle column represents medial TFCF, and the right column represents lateral TFCF. The central row highlights the differences between the means (male−female). Grey areas with corresponding *p*-values from SPM graphs indicate significant and clinically relevant differences.

**Figure 9 bioengineering-10-00179-f009:**
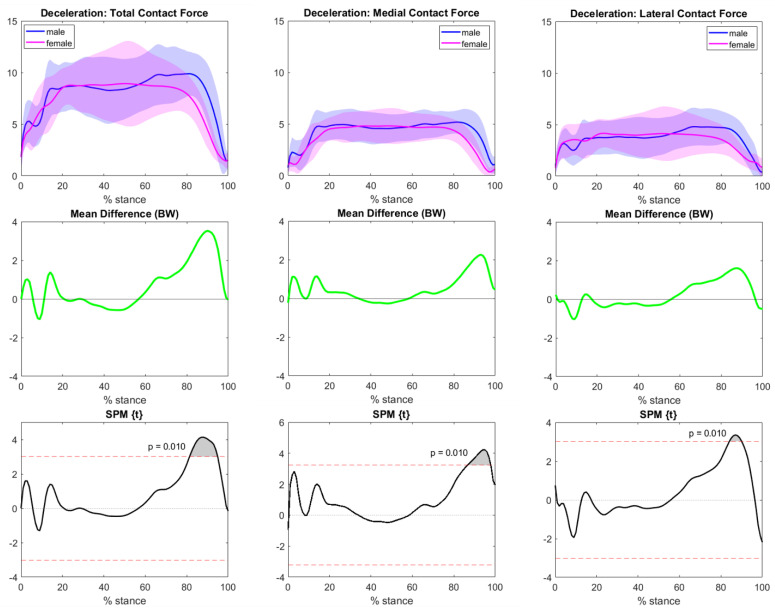
Deceleration TFCF (normalized to body weight and represented as mean and standard deviation) across the task stance in both the male (blue) and female (magenta) groups. The left column represents total TFCF, the middle column represents medial TFCF, and the right column represents lateral TFCF. The central row highlights the differences between the means (male−female). Grey areas with corresponding *p*-values from SPM graphs indicate significant and clinically relevant differences.

**Table 1 bioengineering-10-00179-t001:** General characteristics of analyzed groups (mean and standard deviation). ^+^ and ^#^ symbols highlight significant differences between analyzed groups (*p* < 0.001).

	Total	Male	Female	Varus	Valgus
Numerosity	31	17	14	24	7
Sex (m/f)	17/14	17/0	0/14	12/12	5/2
Age (y)	23.1 ± 4.1	24.2 ± 3.5	21.8 ± 4.4	23.5 ± 4.2	21.6 ± 3.7
Body Mass (kg)	70.0 ± 13.9	79.9 ± 11.0 ^+^	57.9 ± 3.5 ^+^	69.0 ± 14.8	73.4 ± 10.3
Height (cm)	175.4 ± 10.0	181.9 ± 7.0 ^+^	167.5 ± 6.8 ^+^	174.3 ± 10.2	179.0 ± 8.8
BMI (kg/m^2^)	22.55 ± 2.62	24.1 ± 2.43 ^+^	20.68 ± 1.28 ^+^	22.45 ± 2.63	22.91 ± 2.72
Knee Alignment (°)	2.03 ± 2.68	2.21 ± 2.9	1.8 ± 2.48	3.09 ± 1.88 ^#^	−1.63 ± 1.52 ^#^

**Table 2 bioengineering-10-00179-t002:** Synoptic table with peaks and task averages (mean and standard deviation) for total, medial, and lateral TFCF in BW.

	*90° COD*			*DEC*		
*Parameter*	**Total**	**Medial**	**Lateral**	**Total**	**Medial**	**Lateral**
*Peak TFCF (BW)*						
*All*	12.81 ± 2.17	6.73 ± 1.24	6.74 ± 1.68	11.54 ± 2.59	6.23 ± 1.2	5.63 ± 1.66
*Male*	13.02 ± 1.67	6.9 ± 1.03	6.87 ± 1.48	11.97 ± 2.3	6.5 ± 1.16	5.84 ± 1.3
*Female*	12.54 ± 2.69	6.51 ± 1.48	6.58 ± 1.94	11.02 ± 2.92	5.91 ± 1.21	5.37 ± 2.04
*Average TFCF (BW)*						
*All*	7.89 ± 1.33	3.8 ± 0.76	4.09 ± 1.23	7.6 ± 1.7	4.07 ± 0.82	3.59 ± 1.28
*Male*	8.35 ± 1.08	4 ± 0.59	4.35 ± 1.05	7.97 ± 1.6	4.29 ± 0.82	3.68 ± 1.21
*Female*	7.34 ± 1.44	3.56 ± 0.88	3.77 ± 1.39	7.28 ± 1.79	3.8 ± 0.76	3.48 ± 1.41

## Data Availability

The data presented in this study are available on request from the corresponding author. The data are not publicly available due to privacy restrictions.

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
