# Peer review of "Knee Joint Contact Forces during High-Risk Dynamic Tasks: 90° Change of Direction and Deceleration Movements"

_bioengineering, 2023, doi:10.3390/bioengineering10020179_

Round 1

Reviewer 1 Report

First, I would like to congratulate the excellent work that was presented. I understand how laborious it is to prepare a manuscript for publication and we must give proper credit. Congratulations.

You've done a good job. However, I have a few questions to ask.

1.     Lines 33-34: I would suggest adding a reference in this sentence.

2.     Line 36: Please, change the word “more” and add some more examples.

3.     Lines 35-38: Please provide reasons to support what is written in this sentence. Why major risks of occurrence in the young population?

4.     Line 39: What kind of injury? Please provide some examples.

5.     Line 43: What is a high-impact sport?

6.     Line 48: Please change the word "gender" to "sex". Make sure the word is changed throughout the text.

7.     Lines 47-52: Please, the sentence is confusing. It is extensive and contains a lot of information. I suggest splitting it into 2 or 3 parts.

8.     Line 57: Please provide a brief explanation of “articulating compressive forces”.

9.     Lines 82-84: Please, provide a hypothesis.

10.  Line 173: Please, of the 31 players included in the analysis, how many were elite, and how many were recreational (including sex)? In addition, initially, 34 athletes were selected and the final analysis was performed with 34. What was the reason for not having performed the analysis on 3 athletes?

11.  Line 172: All analyzes were performed comparing the entire group. However, from my point of view, there might be differences when you compare elite athletes with recreational athletes. The elite group receives specialized training (which can minimize the possibility of injury) and trains at least 5 times a week. On the other hand, the recreational group does not receive such training and practices sports, on average, twice a week. I would like to see a separate comparison between groups and sex. Please state your opinion on a such comment I made.

12.  I suggest rewriting the conclusion. The information presented in this section has been written previously.

Overall, this opinion article was well written. I was happy to be able to contribute some comments to this paper.

Author Response

Response to Reviewer 1 Comments

We really want to thank the reviewer 1 for the precious suggestions and comments, which have allowed to improve the general quality of the manuscript. Hereinafter, there are the answers to the different points of revision.

 Point 1: Lines 33-34. I would suggest adding a reference in this sentence.

 Response 1: Lines 34-35. We citate two studies among the references.

Point 2: Line 36. Please, change the word “more” and add some more examples.

Response 2: Lines 35-37. We eliminated the word “more” and rephrased the sentence in:

“Health benefits related to sport practice across the lifespan include – but are not limited at – good cardiovascular functionality maintenance, osteogenic homeostasis preservation, weight management, psychologic wellness, and social skills development [1–6].”

We added a reference to the manuscript (Eime, et al. A Systematic Review of the Psychological and Social Benefits of Participation in Sport for Children and Adolescents: Informing Development of a Conceptual Model of Health through Sport. Int J Behav Nutr Phys Act 2013, 10, 98, doi:10.1186/1479-5868-10-98).

Point 3: Lines 35-38. Please provide reasons to support what is written in this sentence. Why major risks of occurrence in the young population?

Response 3: Line 37-39. We thank the reviewer for the question. The likelihood of sport-related musculoskeletal injuries rises as the athlete-exposure time (to practice or game) grows. In last years, the number of sports participants among adolescents has risen, as well as time dedicated to intense sport practice, leading to a rise of injuries in youth and young population. In order to better explain the issue, we added a reference (Prieto-González, et Al, J. Epidemiology of Sports-Related Injuries and Associated Risk Factors in Adolescent Athletes: An Injury Surveillance. Int J Environ Res Public Health 2021, 18, 4857, doi:10.3390/ijerph18094857) and we revised sentences from line 37 to 42 in:

“Nevertheless, it is important to state that sports practice exposure is inextricably linked with the appearance of injuries [7], resulting in major risks of injury occurrence in young population, which have been increasingly dedicating to intense physical activities [5,7]. Sport-related musculoskeletal injuries – such as anterior cruciate ligament (ACL) rupture, chondral damage, meniscal tears, bone fractures – constitute a consistent burden for society. Injured individuals can require long time in rehabilitation or may even suffer long-term consequences on articulations’ health and function, as occurs in case of the onset of early post-traumatic knee osteoarthritis (OA) [8–10].”.

Point 4: Lines 39. What kind of injury? Please provide some examples.

Response 4: We added some examples of musculoskeletal injuries due to sport practice at line 40:

“– such as anterior cruciate ligament (ACL) rupture, chondral damage, meniscal tears, bone fractures –”

Point 5: Lines 39. What kind is a high-impact sport?

Response 5: We thank the reviewer for the question. We intended with the term “high-impact sports” activities in which high forces are generated. However, for sake of clarity, we changed it in “high-dynamics sports” and revised sentence from line 42 to 44 in:

“Traumatic joint injury has been recognized as a strong risk factor for incidence of OA [11], as well as participation at high-dynamics sports where vigorous movements – such as jumping, tackling and pivoting maneuvers – must be performed [11,12].”

Point 6: Lines 48. Please change the word "gender" to "sex". Make sure the word is changed throughout the text.

Response 6: We thank the reviewer for the suggestion. We substituted the word “gender” with “sex” throughout the manuscript (lines 21, 25, 47, 81, 169, 178, 253, 267, 361, 366, 367).

Point 7: Lines 47-52. Please, the sentence is confusing. It is extensive and contains a lot of information. I suggest splitting it into 2 or 3 parts.

Response 7: We thank the reviewer for the suggestion. We divided the sentence from line 48-54 in two parts:

“Whilst high-dynamics sport participation and traumatic joint injuries constitute modifiable risk factors in OA incidence, sex is an endogenous factor, which has been proven to have an evidence-based influence on occurrence of musculoskeletal injuries [15]. For example, female athletes are reported to be more likely subjected to bone stress injuries, in which higher rates of loading represent a risk factor [15], or to ACL tears, where tibiofemoral compression forces play a decisive role in combination with torque joint moments [16,17].”

Point 8: Lines 57. Please, provide a brief explanation of “articulating compressive forces”.

Response 8: Line 58. We substituted “articular compressive forces” statement with “knee tibiofemoral contact forces” for more clarity.

Point 9: Lines 82-84. Please, provide a hypothesis.

Response 9: We substituted the sentence at lines 83-85 with:

“We hypothesized that the exploitation of a generic musculoskeletal model with customable varus-valgus knee alignment could better represent tibiofemoral loadings distribution and could emphasize sexual and anatomical differences.”

Point 10: Line 173: Please, of the 31 players included in the analysis, how many were elite, and how many were recreational (including sex)? In addition, initially, 34 athletes were selected and the final analysis was performed with 34. What was the reason for not having performed the analysis on 3 athletes?

Response 10: In the analysis were included 27 elite athletes (13 males, 14 females) and 4 recreational (4 males). The reason of not using 3 individuals’ data from the original 34 recruitment group is due to loss of data because of exporting issues. We revised Data Processing part in Materials and Methods section and add the following sentence at line 121:

“Because of exporting issues, three individuals’ data were discarded; eventually, thirty-one trials were selected for the analysis (17 males, 14 females).”

Point 11: Line 172: All analyzes were performed comparing the entire group. However, from my point of view, there might be differences when you compare elite athletes with recreational athletes. The elite group receives specialized training (which can minimize the possibility of injury) and trains at least 5 times a week. On the other hand, the recreational group does not receive such training and practices sports, on average, twice a week. I would like to see a separate comparison between groups and sex. Please state your opinion on a such comment I made.

Response 11: We totally agree with the comment of the reviewer on the fact that elite and recreational athletes could be subjected to different specialized trainings which could reduce the possibility of injury. In this perspective, comparison of such two groups would undoubtably give some interesting information. However, our study group was not equally distributed between elite (27 individuals) and recreational (4 individuals) athletes. Such difference in population distribution would invalidate any statistical comparative analysis and could give a misrepresentation of the output. It is also true that we don’t know if nor how often elite athletes’ teams perform injury prevention trainings, which are indeed the major responsible of an effective biomechanics alteration and injury occurrence reduction. Therefore, we added the following sentence at bottom of Discussion section (line 410):

”Finally, it has to be mentioned that the here-presented results are reported considering elite and recreational athletes all together. Elite athletes, characterized in general by a higher amount of time expended in training, could have received specialized training by team coaches aiming to minimize injury occurrence. However, the number of individuals included in the two groups was very dissimilar (27 elite athletes, 4 recreational athletes), so any statistical comparative analysis aiming to highlight dynamics differences lacked in significance. Comparing differences in tibiofemoral forces generated by elite and recreational groups with sex distinction would have provided a more comprehensive overview on task performance mechanism.”.

Point 12: I suggest rewriting the conclusion. The information presented in this section has been written previously.

Response 12: We thank the reviewer for the suggestion. We agree that Conclusion presents already written information. Even though, we think that a short recap could facilitate the undestanding of take-home message, since Discussion section was long and contained a lot of information. However, we revised the Conclusion section (lines 412-428) and add some future perspective considerations:

“In this study, we realized a multibody musculoskeletal model with customizable knee frontal alignment to estimate tibiofemoral contact forces during the execution of high-dynamics movements, with the perspective to objectify movement biomechanics, to better address injury prevention and to support clinical practice in rehabilitation.

As regards 90° change of direction and deceleration tasks, results confirmed that a higher degree of varus knee is associated to bigger medial contact forces. Inter-compartment forces were more similar in sidestepping than in deceleration and the initial part of the movements provided the major risk of injury because of the inherent instability of the articulation. Since loading imbalance on knee compartments determines an overstress of articular soft tissues (ligaments and cartilage) which could lead to serious injuries, technique and rehabilitative exercises should pay specific attention in training athletes’ neuromuscular response to deal with the first instants of deceleration and sidestepping braking phases.

As a further result of the study, we highlighted that men in general were prone to develop major internal forces during the execution of sport activity, albeit in general lateral forces were not higher in male individuals respect to female athletes, as the propension of valgus-loading-relate injuries of the latter could seem indicate.

The biomechanical simulation of high dynamics tasks, often occurring in game situations, is of paramount importance to increase knowledge about the underlying mechanisms of injury. Understanding which are the major contributors – among all exogenous and endogenous factors – impacting on the execution of a certain task would have the potentiality to elaborate specific training programs aiming to prevent and reduce sport injury occurrence, thereby relieving the burden on national Healthcare Systems.

Future research should focus on developing more and more accurate tools to reliably estimate sport-related injury biomarkers. On the other side, exploiting novel technologies capable to record motion outside the lab would allow a comprehensive evaluation of high-dynamics tasks situations: in this perspective, the recent trend in using inertial measurement units for motion analysis [59] could open interesting opportunities.”

We add the following reference: Konrath, et al. Estimation of the Knee Adduction Moment and Joint Contact Force during Daily Living Activities Using Inertial Motion Capture. Sensors (Basel) 2019, 19, E1681, doi:10.3390/s19071681.

Reviewer 2 Report

This aim of this study is to present a musculoskeletal model for varus and valgus knees with adjustable knee frontal alignment to estimate tibiofemoral contact forces during 90 degree sidestepping and deceleration tasks. I think this study is merit and the presented results is of great importance in the field of injury biomechanics. In order to clarify the procedure, there are only several questions I would like to ask and there are also some typo mistakes:

1) Figures 2 and 3 show relative change in force. Can you explain relative with respect to what value?

2)  In this study, the authors claimed they used an OpenSim musculoskeletal model. No results related to muscle forces or activations were reported. If the aim of this study is to compute tibiofemoral contact forces in the knee, there is no need to apply a musculoskeletal model, a skeletal model can provide such results. Would you please explain why you applied a musculoskeletal model for just computing bone on bone contact forces?

3) In page 6, caption of Figure. 2, there is a typo mistake. Please change “Varus-Varus alignment” to “Valgus-Varus alignment”.

4) Please use the same order of panels in Figure 2 for Figure 3. Medial in the left column and lateral in the right column.

5) Please explain how you compute the curves in the first row of Figures 4 and 5. Are they the average of all trials for men and women with varus and valgus knee?

6) The second panel from top in Figure 6 shows the medial force minus lateral force; while in the caption, it is said (lateral-medial), please revise it. The same thing is true for the Figure 7, second panel from top.

Author Response

Response to Reviewer 2 Comments

We really want to thank the reviewer 2 for the precious comments, which have allowed to improve the general quality of the manuscript. Hereinafter, there are the answers to the different points of revision.

 Point 1: Figures 2 and 3 show relative change in force. Can you explain relative with respect to what value?

Response 1: Force differences showed in figure 2 and 3 were the difference measured by aligned-knee models respect to neutral-knee models (which do not consider valgus-varus alignement of the knee). We revised caption of figure 2 adding the following sentence (line 196):

“Change (Δmean) is intended as the difference between task-average forces measured by aligned-model respect to neutral-model; every dot represents one individual case.”

We add the following sentence to caption of figure 3 (line 200):

“Change (Δmean) is intended as the difference between peak forces measured by aligned-model respect to neutral-model; every dot represents one individual case.”

Point 2: In this study, the authors claimed they used an OpenSim musculoskeletal model. No results related to muscle forces or activations were reported. If the aim of this study is to compute tibiofemoral contact forces in the knee, there is no need to apply a musculoskeletal model, a skeletal model can provide such results. Would you please explain why you applied a musculoskeletal model for just computing bone on bone contact forces?

Response 2: We thank the reviewer for the question. It is true that in this study we didn’t show muscular output. However, their computation is of paramount importance to correctly estimate articular internal forces when multi-body modelling simulations are conducted. Briefly, running a standard simulation of biomechanics in OpenSim consists in running Inverse Kinematics (IK) analysis, Inverse Dynamics (ID) analysis, Static Optimization (SO) analysis, and finally Joint Reaction (JR) analysis in sequence. IK allows to reconstruct the movement of body segments of the skeletal model according to motion capture data input and certain constraints imposed to the model. ID determines the generalized forces (e.g., net forces and torques) at each joint responsible for a given movement, based on equation of motion F=ma; it must be specified that ID only considers external forces (ground reaction forces and inertial forces) to compute its generalized forces, excluding internal forces from the computation. SO allows to resolve net joint moments into individual muscle forces at each instant in time, applying certain criteria to resolve muscles redundancy problem. JR takes in consideration all loads acting on the model – either external (i.e., ground reaction and inertial forces) and internal (muscles compression activity between joints) – to correctly estimate the magnitude of articular forces. Several studies have highlighted that the effect of muscles is preponderant in determining the effective loading acting on a joint, especially when high dynamics tasks are performed: as an example, Saxby and colleagues (reference 21 in our manuscript) stated that muscles contributed for more of 75% of the total to tibiofemoral contact forces during running and sidestepping.

Other tools, beside multi-body modelling, are available to estimate tibiofemoral contact forces. For example, it is possible to run kinematic-driven Finite Element Modeling (FEM) simulations which allow to estimate contact forces without the necessity to know muscles activity. However, this kind of simulations are time-greedy and require a deep knowledge of the mechanical properties of the joint tissue (bone, cartilage, ligaments. menisci). Moreover, a kinematic-driven FEM simulation requires very accurate kinematics: this is something that can barely achieved by means of particular motion capture techniques such as X-ray stereophotogrammetry or dynamic MRI, which present either problems of invasiveness, costs or movement-constrains. Marker-based motion capture does not allow to achieve accurate estimation of certain knee movement degree of freedom such as varus/valgus rotation or tibiofemoral vertical displacement during the movement, thus it is usually paired with multi-body musculoskeletal modelling.

Point 3: In page 6, caption of Figure. 2, there is a typo mistake. Please change “Varus-Varus alignment” to “Valgus-Varus alignment”.

Response 3: We thank the reviewer for the correction. We substituted “Varus-Varus alignment” in captions of figure 2 (line 194) and 3 (line 198) with “Valgus-Varus alignment” statement.

Point 4: Please use the same order of panels in Figure 2 for Figure 3. Medial in the left column and lateral in the right column.

Response 4: We thank the reviewer for the correction. We fixed the panels of Figure 3 (line 197). Now both Figure 2 and 3 present medial forces on the left and lateral forces on the right.

Point 5: Please explain how you compute the curves in the first row of Figures 4 and 5. Are they the average of all trials for men and women with varus and valgus knee?

Response 5: Curves of the first row of Figures 4 and 5 are the average of all analyzed subject (i.e., men, women, varus, valgus knee groups all together). We revised the first sentence of caption of Figure 4 (line 209) in:

“Sidestepping TFCF (normalized to body weight and represented as mean and standard deviation) of all the individuals across the task stance, compared between the neutral (red) and aligned (blue) knee”

We revised the first sentence of caption of Figure 5 (line 210) in:

“Deceleration TFCF (normalized to body weight and represented as mean and standard deviation) of all the individuals across the task stance, compared between the neutral (red) and aligned (blue) knee”

Moreover, we added the following statement at the beginning of Neutral vs aligned models TFCF subsection in Results (line 202):

“Total, medial, and lateral TFCF estimated by neutral-knee and aligned-knee models were averaged considering all individual trials, and SPM analysis between the two modelling output was conducted”

Point 6: The second panel from top in Figure 6 shows the medial force minus lateral force; while in the caption, it is said (lateral-medial), please revise it. The same thing is true for the Figure 7, second panel from top.

Response 6: We thank the reviewer for the correction. We revised captions of Figures 6 and 7 substituting “lateral-medial” with “medial-lateral” statement.

Round 2

Reviewer 1 Report

Very good job!